# Auto-Ensemble Structure Learning of Large Gaussian Bayesian Networks

## Abstract

Learning the structure of Bayesian networks (BNs) from data is challenging, especially for datasets involving a large number of variables. The recently proposed divide-and-conquer (D&D) strategies present a promising approach for learning large BNs. However, they still face a main issue of unstable learning accuracy across subproblems. In this work, we introduce the idea of employing structure learning ensemble (SLE), which combines multiple BN structure learning algorithms together, to consistently achieve high learning accuracy across various problems. We further propose an automatic approach called Auto-SLE for constructing near-optimal SLEs, addressing the challenge of manually designing effective SLEs. The automatically constructed SLE is then integrated into a D&D framework. Extensive experiments firmly show the superiority of our method over existing methods in learning large BNs, achieving accuracy improvement usually by 30%~225% on datasets involving 10,000 variables. Furthermore, our method generalizes very well to datasets with many more variables and different network characteristics than those present in the training data for constructing the SLE. These results indicate the significant potential of employing automatic construction of SLEs for BN structure learning.

## 1 Introduction

Learning the structure of Bayesian networks (BNs) (Pearl, 1985) from data has attracted much research interest, due to its wide applications in machine learning, statistical modeling, and causal inference (Pearl, 1988; Kitson et al., 2023). Various methods have been proposed to tackle this problem, including constraint-based methods (Colombo et al., 2014), score-based methods (Ramsey et al., 2017), and hybrid methods (Tsamardinos et al., 2006). However, most previous studies primarily dealt with a relatively small number of variables. For example, the *bnlearn* repository (Scutari, 2010), which is widely used in the literature, contains mostly networks with only a few dozen nodes (variables). In contrast, in real-world applications such as alarm events analysis (Cai et al., 2022), MRI image interpretation (Ramsey et al., 2017), and human genome analysis (Schaffter et al., 2011), it is common to generate and collect data from thousands of variables and beyond. Unfortunately, as the number of variables increases, many of the existing methods would slow down dramatically and become much less accurate (Zhu et al., 2021).

Recently, Gu and Zhou (2020) introduced a divide-and-conquer (D&D) framework named partition-estimation-fusion (PEF) for the structure learning of large BNs. Specifically, PEF consists of three steps: partitioning nodes into clusters (partition), learning a subgraph on each cluster (estimation), and fusing all subgraphs into a single BN (fusion). It is noteworthy that PEF is flexible as any existing structure learning algorithm can be used in the estimation step. Additionally, due to the smaller number of nodes in each cluster compared to the total number of nodes, the overall structure learning is accelerated. Moreover, the structure learning processes for different subproblems can be parallelized in a straightforward way, leading to further improvement in computational efficiency.

However, despite the evident advantages provided by PEF, it still faces a main issue of unstable structure learning accuracy across subproblems. The root cause for this is that the partition step of PEF tends to yield subproblems with significantly different characteristics, e.g., varying node numbers. When a single structure learning algorithm is used to solve all subproblems, as in existing PEF-based methods (Gu & Zhou, 2020), achieving stable learning accuracy across different

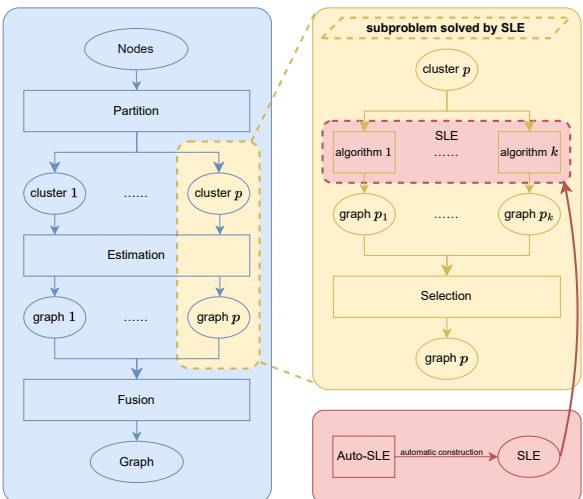

Figure 1: An overview of P/SLE. The SLE is automatically constructed by Auto-SLE and integrated into the estimation step of PEF.

subproblems becomes challenging. In fact, even for the same algorithm, different hyperparameter values can lead to significant variations in its behavior, thereby affecting its suitability for solving specific subproblems. For instance, when employing the well-known fast greedy equivalence search (fGES) (Ramsey et al., 2017) algorithm with Bayesian information criterion (BIC) as the score function, its penalty coefficient should ideally increase with the sparsity of the underlying BN. However, given a BN structure learning problem, determining the optimal penalty coefficient in advance is difficult as the underlying BN is unknown.

Inspired by the success of ensemble methods like AutoAttack (Croce & Hein, 2020) in the field of adversarial robustness, which utilizes an attack ensemble to achieve more reliable robustness evaluation compared to individual attacks, we introduce the idea of employing structure learning ensemble (SLE) to achieve stable learning accuracy in BN structure learning. Specifically, a SLE comprises several structure learning algorithms, dubbed member algorithms. When applied to a BN structure learning problem, a SLE runs its member algorithms individually and chooses the best of their outputs. Similar to how AutoAttack integrates diverse attacks, a high-performing SLE should consist of complementary member algorithms that excel at solving different types of problems. However, *in contrast to AutoAttack where the attacks are manually designed and selected, we propose to automatically construct SLEs*. This can significantly reduce the reliance on human expertise and effort, as manually constructing SLEs typically requires domain experts to explore the vast design space of SLEs, which can be both laborious and intricate.

Specifically, we first formulate the problem of automatic SLE construction, and then present Auto-SLE, a simple yet effective approach to address this problem. The approach is implemented with a design space containing several candidate algorithms; each algorithm has a set of hyperparameters. To construct a SLE, Auto-SLE starts from an empty ensemble, and repeatedly adds the algorithm and its associated hyperparameter values to the ensemble such that the ensemble improvement is maximized. In addition to its conceptual simplicity, Auto-SLE is appealing also because it yields SLEs provably within a constant factor of optimal for the training problem set. To achieve our goal of constructing a single SLE that can generalize well across network sizes and characteristics, we utilize a training set comprising diverse networks of varying sizes. Finally, the constructed SLE is integrated into the PEF framework, resulting in P/SLE (see Figure 1 for an overview).

From extensive comparisons with the state-of-the-art baselines, it is found that P/SLE consistently achieves significantly higher accuracy in learning large BNs, and its superiority becomes even more pronounced as the network size further increases. On datasets involving 10,000 variables, P/SLE typically achieves accuracy improvement by 30%∼225%. Further experiments show that P/SLE, without any additional tuning or adaptation, generalizes well to datasets with much larger number (e.g., 30,000) of variables and possessing different network characteristics than the training data.

It is worth mentioning that Auto-SLE itself is a general SLE construction approach that can be applied to various types of BNs (Gaussian and non-Gaussian) and even other types of causal models. In this work, we focus on Gaussian BNs, which are arguably the most widely studied type of BNs and have a rich set of baselines, allowing us to thoroughly assess the potential of Auto-SLE. Besides, it should be noted that PEF-based methods, such as P/SLE, are most suitable for learning BNs with a block structure to some extent, meaning the connections between subgraphs are relatively weak. It is quite common for a large network to exhibit such a block structure, due to the underlying heterogeneity among the nodes (Holland et al., 1983; Airoldi et al., 2008; Abbe et al., 2015). On the other hand, our results also indicate that, even for large BNs without a block structure, P/SLE can still achieve competitive learning accuracy. The promising performance of P/SLE not only demonstrates the potential of applying SLEs to BN structure learning, a largely unexplored area, but also highlights the effectiveness of Auto-SLE as a simple and easy-to-use approach for constructing SLEs. This is expected to further facilitate the advancement of SLEs in this field.

## 2 PRELIMINARIES AND RELATED WORK

### 2.1 THE BN STRUCTURE LEARNING PROBLEM

The structure of a BN for $d$ random variables $X_1, ..., X_d$ is represented by a directed acyclic graph (DAG), denoted as $\mathcal{G} = (V, E)$. Here, $V = \{1, ..., d\}$ is the set of nodes corresponding to the random variables and $E = \{(j, i) \in V \times V : j \to i\}$ is the directed edge set. Define $V_{\text{pa}(i)} = \{j \in V : (j, i) \in E\}$ as the parent node set of node $i$, and $X_{\text{pa}(i)}$ as the set of corresponding random variables. The joint probability density function $f$ of $X_1, ..., X_d$ is factorized according to the structure of $\mathcal{G}$:

$$f(X_1, X_2, ..., X_d) = \prod_{i=1}^{d} f\left(X_i | X_{\text{pa}(i)}\right), \qquad (1)$$

where $f\left(X_i | X_{\text{pa}(i)}\right)$ is the conditional probability density of $X_i$ given $X_{\text{pa}(i)}$. In this work, we focus on Gaussian BNs for continuous data. Specifically, the conditional distributions are specified by the following linear structural equation model:

$$X_i = \phi\left(X_{\text{pa}(i)}\right) + \varepsilon_i, \quad i = 1, ..., d, \qquad (2)$$

where $\phi(\cdot)$ denotes a linear function and $\varepsilon_i \sim \mathcal{N}\left(0, \sigma_j^2\right)$. Suppose we have obtained $m$ iid observations of $X_1, \ldots, X_d$, denoted as $D \in \mathbb{R}^{m \times d}$. Given $D$, the goal is to learn a DAG structure $\mathcal{G} = (V, E)$ that accurately reflects the conditional dependencies among $X_1, \ldots, X_d$. In practice, accuracy metrics such as F1 score and structural Hamming distance (SHD) are typically used to assess the quality of the learned BN.

### 2.2 RELATED WORK

The BN structure learning problem has been proven to be NP-hard (Chickering et al., 2004), leading to main research efforts on developing approximation methods to solve it. These methods can be broadly classified into constraint-based, score-based, and hybrid methods. Constraint-based methods, such as PC (Spirtes et al., 2000), MMPC (Tsamardinos et al., 2003a), and PC-Stable (Colombo et al., 2014), use conditional independence tests on observations to identify relationships among variables. In comparison, score-based methods explore the space of DAGs or Markov equivalence classes (MECs) using search heuristics such as tabu search (TS) (Bouckaert, 1995), genetic algorithm (GA) (Larranaga et al., 1996), and greedy search (Chickering, 2002). These methods also employ score functions, such as BDeu (Akaike, 1974), BIC (Schwarz, 1978), and K2 (Cooper & Herskovits, 1992), to guide the search. It is worth mentioning a recent research line of score-based methods including NOTEARS (Zheng et al., 2018) and LEAST (Zhu et al., 2021) that reformulate the structure learning as a continuous optimization problem. Finally, hybrid methods integrate both constraint-based and score-based techniques. For example, MMHC (Tsamardinos et al., 2006) uses MMPC to build the graph skeleton and utilizes TS to determine the final BN.

However, as the number of variables increases, many of the existing methods would slow down dramatically and become much less accurate (Zhu et al., 2021). Actually, based on our preliminary testing of 15 existing methods, fGES (Ramsey et al., 2017), which is a variant of the greedy search

algorithm (Chickering, 2002), and PC-Stable (Colombo et al., 2014), are the only methods capable of maintaining relatively stable learning accuracy (e.g., achieving F1 score of around 0.5) within reasonable time when the variable number reaches 1000. Moreover, when the number of variables reaches 10,000, all methods, even after running for a quite long period of time (e.g., 24 hours), are unable to output solutions.

**PEF Framework** To address the challenge of learning large BNs, Gu and Zhou (2020) introduced a partition-estimation-fusion (PEF) framework. It comprises the following three steps.

- **Partition**: The $d$ nodes are partitioned into clusters with a hierarchical clustering algorithm.
- **Estimation**: An existing structure learning algorithm is applied to estimate a subgraph on each cluster of nodes.
- **Fusion**: Merge estimated subgraphs into one DAG containing all the $d$ nodes.

The details of PEF can be found in Appendix A. While PEF has dramatically enhanced the capabilities of handling large BNs, it still faces a main issue of unstable structure learning accuracy across subproblems. In the following we will describe the use of automatically constructed SLE in the estimation step to address this issue.

## 3 AUTOMATIC CONSTRUCTION OF SLEs

### 3.1 PROBLEM FORMULATION

We first formulate the SLE construction problem. Formally, a SLE with $k$ member algorithms is denoted as $\mathbb{A} = \{\theta_1, \ldots, \theta_k\}$, where $\theta_i$ represents the $i$-th member algorithm of $\mathbb{A}$. Let $T = \{D_1, D_2, \ldots\}$ denote a training problem set, where each $D_i$ represents a BN structure learning problem with known ground truth. When using $\mathbb{A}$ to solve a problem $D \in T$, one straightforward strategy is to run all member algorithms of $\mathbb{A}$ individually in parallel, and the best solution among all the found solutions in terms of a quality measure $Q$ (e.g., F1 score) is returned. Let $Q(\mathbb{A}, D)$ and $Q(\theta_i, D)$ denote the performance of $\mathbb{A}$ and $\theta_i$ on $D$ in terms of $Q$, respectively. Without loss of generality, we assume a larger value is better for $Q$. Then we have:

$$Q(\mathbb{A}, D) = \max_{\theta_i \in \mathbb{A}} Q(\theta_i, D). \tag{3}$$

Then the performance of $\mathbb{A}$ on $T$ in terms of $Q$, denoted as $Q(\mathbb{A}, T)$, is the average value of $\mathbb{A}$'s performance on the training problems in $T$ ($Q(\mathbb{A}, T) = 0$ when $\mathbb{A}$ is empty):

$$Q(\mathbb{A}, T) = \frac{1}{|T|} \sum_{D \in T} Q(\mathbb{A}, D). \tag{4}$$

For the automatic construction of $\mathbb{A}$, the member algorithms of $\mathbb{A}$ are not manually determined but automatically selected from an algorithm configuration space $\mathbf{\Theta}$. Specifically, suppose we have a candidate algorithm pool $\{\mathcal{A}_1, \ldots, \mathcal{A}_n\}$, which can be constructed by collecting existing algorithms. Each candidate algorithm has some hyperparameters. Let $\Theta_i$ denote the hyperparameter configuration space of $\mathcal{A}_i$, where a configuration $\theta \in \Theta_i$ refers to a setting of $\mathcal{A}_i$'s hyperparameters, such that its behaviors is completely specified. Then, the algorithm configuration space $\mathbf{\Theta}$ is defined as $\mathbf{\Theta} = \Theta_1 \cup \Theta_2 \cdots \cup \Theta_n$, and each $\theta \in \mathbf{\Theta}$ represents a specific candidate algorithm along with the specific values of its hyperparameters. As presented in Definition 1, the SLE construction problem is to select $k$ member algorithms from $\mathbf{\Theta}$ to form a SLE $\mathbb{A}^*$, such that its performance on the training set $T$ in terms of $Q$ is maximized.

**Definition 1.** *Given $T, Q, \mathbf{\Theta}$, and $k$, the SLE construction problem is to find $\mathbb{A}^* = \{\theta_1^* \ldots \theta_k^*\}$ that maximizes $Q(\mathbb{A}^*, T)$, s.t. $\theta_i^* \in \mathbf{\Theta}$ for $i = 1 \ldots k$.*

### 3.2 AUTO-SLE: A GREEDY APPROACH

We now introduce Auto-SLE, a simple yet effective approach for automatically constructing SLEs. As shown in Algorithm 1, Auto-SLE starts with an empty ensemble $\mathbb{A}$ (line 1) and finds the candidate algorithm and its hyperparameter values denoted as $\theta^\circ$ that, if included in $\mathbb{A}$, maximizes ensemble

---

**Algorithm 1:** Auto-SLE

---

**Input:** quality measure $Q$, training set $T$, algorithm configuration space $\boldsymbol{\Theta}$, ensemble size $k$
**Output:** $\mathbb{A}$

1   $\mathbb{A} \leftarrow \varnothing, i \leftarrow 1$;
2   **while** $i \leq k$ **do**
3      $\theta^\circ \leftarrow \arg\max_{\theta \in \boldsymbol{\Theta}} Q\left(\mathbb{A} \cup \{\theta\}, T\right) - Q\left(\mathbb{A}, T\right)$;
4      **if** $Q\left(\mathbb{A} \cup \{\theta^\circ\}, T\right) = Q\left(\mathbb{A}, T\right)$ **then return** $\mathbb{A}$;
5      $\mathbb{A} \leftarrow \mathbb{A} \cup \{\theta^\circ\}, i \leftarrow i + 1$;
6   **end**
7   **return** $\mathbb{A}$

---

improvement in terms of $Q$ (line 3). Ties are broken arbitrarily here. After this, $\theta^\circ$ is subject to the following procedure: if adding it to $\mathbb{A}$ does not improve performance, which means the construction process has converged, Auto-SLE will terminate and return $\mathbb{A}$ (line 4); otherwise $\theta^\circ$ is added to $\mathbb{A}$ (line 5). The above process will be repeated until $k$ member algorithms have been found (line 2).

Let $\Delta(\theta|\mathbb{A}) = Q\left(\mathbb{A} \cup \{\theta\}, T\right) - Q\left(\mathbb{A}, T\right)$ denote the performance improvement brought by adding $\theta$ to $\mathbb{A}$. Noticing that each iteration of Auto-SLE needs to find $\theta^\circ$ that maximizes $\Delta(\theta|\mathbb{A})$, when the algorithm configuration space $\boldsymbol{\Theta}$ is large or even infinite (e.g., candidate algorithms have continuous hyperparameters), using enumeration to find $\theta^\circ$ is impractical. In practice, we employ hyperparameter optimization procedures, such as Bayesian optimization (Lindauer et al., 2022), to approximately maximize $\Delta(\theta|\mathbb{A})$, and the runs of these procedures account for the vast majority of the total computational costs of Auto-SLE.

## 3.3 THEORETICAL JUSTIFICATIONS

We now theoretically analyze the performance of Auto-SLE on a give training problem set. For notational simplicity, henceforth we omit the $T$ in $Q(\cdot, T)$ and directly use $Q(\cdot)$. Our analysis is based on the following key fact that $Q(\cdot)$ is monotone and submodular.

**Fact 1.** *$Q(\cdot)$ is a monotone submodular function, i.e., for any two SLEs $\mathbb{A}, \mathbb{A}' \subset \boldsymbol{\Theta}$ and any $\theta \in \boldsymbol{\Theta}$, it holds that $Q(\mathbb{A}) \leq Q(\mathbb{A} \cup \mathbb{A}')$ and $Q\left(\mathbb{A} \cup \mathbb{A}' \cup \{\theta\}\right) - Q\left(\mathbb{A} \cup \mathbb{A}'\right) \leq Q\left(\mathbb{A} \cup \{\theta\}\right) - Q\left(\mathbb{A}\right)$.*

The proof is straightforward and can be found in Appendix B. Intuitively, $Q$ exhibits a diminishing returns property that the marginal gain of adding $\theta$ diminishes as the ensemble size increases. Based on Fact 1, Theorem 1 holds.

**Theorem 1.** *Using a hyperparameter optimization procedure that, in each iteration of Auto-SLE, returns $\hat{\theta}$ within $\epsilon$-absolute error of the maximum of $\Delta(\theta|\mathbb{A})$, i.e., $\Delta(\hat{\theta}|\mathbb{A}) \geq \Delta(\theta^\circ|\mathbb{A}) - \epsilon$, then the quality $Q(\mathbb{A})$ of the SLE constructed by Auto-SLE is bounded by*

$$Q(\mathbb{A}) \geq (1 - 1/e) \cdot Q(\mathbb{A}^*) - k\epsilon, \tag{5}$$

*where $\mathbb{A}^*$ is the optimal SLE to the SLE construction problem in Definition 1. Alternatively, if $\hat{\theta}$ is within $\epsilon$-relative error of $\Delta(\theta^\circ|\mathbb{A})$, i.e., $\Delta(\hat{\theta}|\mathbb{A}) \geq \Delta(\theta^\circ|\mathbb{A}) \cdot (1 - \epsilon)$, then the quality $Q(\mathbb{A})$ of the SLE constructed by Auto-SLE is bounded by*

$$Q(\mathbb{A}) \geq (1 - 1/e^{1-\epsilon}) \cdot Q(\mathbb{A}^*). \tag{6}$$

The proof (see Appendix B) is a slight extension of the classical derivations in maximizing $\Delta(\theta|\mathbb{A})$ (Nemhauser et al., 1978). Based on Theorem 1, Auto-SLE achieves $(1 - 1/e)$-approximation for the optimal quality when given a perfect hyperparameter optimization procedure with $\epsilon = 0$. Suboptimal hyperparameter optimization procedures result in worse outcomes but small errors $\epsilon$ do not escalate. This is important because with large algorithm configuration space, it cannot be expected for blackbox optimization procedures to find $\theta^\circ$ in realistic time. However, at least for some scenarios with a few parameters, widely-used Bayesian optimization techniques such as SMAC (Lindauer et al., 2022) have empirically been shown to yield performance close to optimal within reasonable time budget.

Note that Theorem 1 only provides a performance guarantee for the training problem set used for constructing the SLE. By employing the same techniques introduced by Liu et al. (2020), we can

recover guarantees for an independent test set that is sampled or generated from the same distribution as the training set, given a sufficiently large training set. Nevertheless, our experimental results (see Section 4.2) demonstrate that the constructed SLE performs well beyond the training set.

### 3.4 APPLYING THE CONSTRUCTED SLE TO A NEW PROBLEM

Importantly, when presented with a testing problem to solve, it cannot be assumed that the ground truth is known. Thus, quality measures that do not require ground truth, such as the BIC score adopted in our experiments, are used to select the best output from the outputs of member algorithms. It is worth mentioning that although this work focuses on utilizing the constructed SLE to enhance PEF, the SLE itself can also serve as a complete and independent BN structure learning method.

## 4 EXPERIMENTS

Extensive experiments are conducted to answer two research questions (RQs).

- **RQ1**: Does the SLE constructed by Auto-SLE enhance PEF in learning large BNs?

- **RQ2**: Can the SLE generalize to larger problem sizes and different network characteristics than that present in the training set?

### 4.1 EXPERIMENTAL SETUP

**Benchmarks and Evaluation Metrics**   We generate diverse and large-scale benchmarks following the approach in (Gu & Zhou, 2020). Specifically, the approach consists of four steps: (i) select an existing network structure and replicates it until a predefined variable number is reached; (ii) connect the replicas by adding $10\%$ of edges between them randomly, while ensuring the final network remains a DAG; (iii) utilize the complete DAG and Eq. (2) to generate observations, where the weights in the linear function and the standard deviations of Gaussian noises are sampled uniformly from $[-1, -0.5] \cup [0.5, 1]$ and $[0, 1]$, respectively; (iv) re-scale the observations such that all data columns have the same mean and standard deviation. We select 10 networks from the *bnlearn* repository (Scutari, 2010) with node numbers ranging from 5 to 441. Based on each of them, we use the above approach to generate testing problems with around 1000 and 10,000 variables. Following (Gu & Zhou, 2020), the sample size $m$ in each testing problem is set to 1000.

The widely-used F1 score and SHD are adopted as the metrics for assessing learning accuracy. In line with previous comparative study (Ramsey et al., 2017), we use two specific variants of F1 score, i.e., F1 Arrowhead ($F1^{\rightarrow}$) that considers direction and F1 Adjacent ($F1^{-}$) that ignores direction, since some methods (PC-Stable and fGES) output MECs of DAGs which may contain edges without directions. Moreover, the wall-clock runtime of the methods is reported. For F1 metrics, a higher value is better; for SHD and runtime, a lower value is better.

**Constructing the SLE with Auto-SLE**   A diverse training problem set is beneficial for constructing a SLE with good performance across problem sizes and network characteristics. Specifically, the training set $T$ comprises 100 problems, with variable numbers ranging from 5 to 1000, generated using the above approach based on a network randomly selected from the 32 networks in *bnlearn*. These problems are exclusively used for constructing the SLE and are independent of the testing problems. For the algorithm configuration space $\Theta$, we consider two algorithms PC-stable (with two hyperparameters) and fGES (with three hyperparameters) as candidate algorithms, which outperform others in our preliminary testing. We use their implementations from the causal discovery tool box *TETRAD* (Ramsey et al., 2018). The sum of $F1^{-}$ and $F1^{\rightarrow}$ is adopted as the quality measure $Q$, and ensemble size $k$ is set to $4$ as running more iterations of Auto-SLE brings a negligible improvement to the SLE's performance on the training set (see Appendix C.3). SMAC (version 3) (Lindauer et al., 2022), a Bayesian optimization tool, is used to maximize $\Delta(\theta|\mathbb{A})$ in each iteration of Auto-SLE, with a time budget 12 hours per run. Consequently, Auto-SLE consumes approximately 48 hours in total to construct the SLE. Then, the SLE is integrated into PEF, resulting in P/SLE, which is evaluated in subsequent experiments without further tuning or adaptation.

Table 1: Results on testing problems with 1000 variables, in terms of F1 Adjacent ($F1^-$), F1 Arrowhead ($F1^\rightarrow$), SHD, and runtime (T). On each network, the mean ± std performance obtained by each method on 10 problems is reported. The best performance in terms of accuracy metrics is marked with an underline, and the performance that is not significantly different from the best performance (according to a Wilcoxon signed-rank test with significance level $p = 0.05$) is indicated in **bold**. Let B be the best performance achieved among the baselines and A be the performance of P/SLE. The improvement (Impro.) ratio is calculated as (A-B)/B for F1 metrics (a higher value is better), and is calculated as (B-A)/B for SHD (a lower value is better).

| Problem ($|V|,|E|$) | | Alarm (1036,1417) | Asia (1000,1100) | Cancer (1000,880) | Child (1000,1375) | Earthquake (1000,880) | Hailfinder (1008,1307) | Healthcare (1001,1416) | Mildew (1015,1468) | Pigs (1323,1954) | Survey (1002,1103) |
|---|---|---|---|---|---|---|---|---|---|---|---|
| P/SLE | $F1^-$ | **0.81±0.02** | **0.96±0.00** | **0.98±0.00** | **0.86±0.01** | **0.98±0.00** | **0.80±0.02** | **0.89±0.01** | **0.68±0.02** | **0.70±0.02** | **0.92±0.01** |
| | $F1^\rightarrow$ | **0.68±0.04** | **0.74±0.01** | **0.94±0.01** | **0.60±0.02** | **0.94±0.01** | **0.61±0.03** | 0.59±0.01 | **0.54±0.03** | **0.61±0.02** | **0.79±0.02** |
| | SHD | **625.7±73.6** | **125.2±15.2** | **43.1±6.9** | **581.1±23.3** | **46.1±8.1** | **571.3±62.7** | **485.0±38.4** | **1135.4±71.8** | **1262.8±84.4** | **309.9±29.8** |
| | T (s) | 7.4±0.2 | 5.7±0.1 | 5.1±0.1 | 8.4±0.3 | 5.1±0.1 | 243.3±249.0 | 6.4±0.2 | 8.7±0.6 | 726.9±1176.6 | 6.0±0.1 |
| P/SLE(D) | $F1^-$ | 0.68±0.01 | 0.78±0.01 | 0.74±0.01 | 0.75±0.01 | 0.74±0.01 | 0.66±0.01 | 0.77±0.01 | 0.56±0.02 | 0.53±0.01 | 0.75±0.01 |
| | $F1^\rightarrow$ | 0.57±0.03 | 0.64±0.01 | 0.73±0.01 | 0.56±0.01 | 0.72±0.01 | 0.57±0.02 | 0.56±0.01 | 0.45±0.03 | 0.47±0.02 | 0.62±0.02 |
| | SHD | 1234.3±77.9 | 710.5±24.7 | 611.1±31.2 | 1055.5±39.4 | 617.5±27.1 | 1158.4±65.2 | 1051.6±31.0 | 1826.2±100.4 | 2442.4±115.6 | 871.6±32.0 |
| | T (s) | 26.0±1.8 | 20.4±3.0 | 17.1±2.7 | 21.6±5.0 | 15.4±2.3 | 366.0±287.4 | 9.7±1.8 | 14.2±2.2 | 5697.5±1870.6 | 7.8±0.1 |
| P/SLE(R) | $F1^-$ | 0.72±0.02 | 0.87±0.00 | 0.77±0.01 | 0.83±0.00 | 0.77±0.01 | 0.66±0.01 | 0.87±0.01 | 0.57±0.02 | 0.58±0.04 | 0.84±0.01 |
| | $F1^\rightarrow$ | 0.42±0.02 | 0.49±0.01 | 0.32±0.03 | 0.51±0.02 | 0.30±0.02 | 0.44±0.02 | 0.44±0.01 | 0.31±0.01 | 0.35±0.03 | 0.46±0.01 |
| | SHD | 1070.1±56.9 | 511.7±17.2 | 697.1±44.0 | 680.7±34.7 | 703.7±34.5 | 1152.3±38.7 | 750.8±31.6 | 1647.1±60.2 | 1934.3±60.6 | 675.9±28.1 |
| | T (s) | 19.6±1.1 | 12.2±1.9 | 11.4±1.9 | 13.7±1.7 | 10.4±1.4 | 33.3±8.3 | 12.9±3.1 | 15.5±1.8 | 5886.7±1671.0 | 8.3±2.0 |
| P/PC-Stable | $F1^-$ | 0.76±0.02 | 0.91±0.00 | 0.85±0.01 | 0.85±0.01 | 0.85±0.01 | 0.71±0.01 | **0.89±0.01** | 0.62±0.02 | 0.62±0.05 | 0.89±0.01 |
| | $F1^\rightarrow$ | 0.48±0.02 | 0.56±0.01 | 0.47±0.03 | 0.52±0.02 | 0.43±0.02 | 0.46±0.01 | 0.46±0.01 | 0.35±0.02 | 0.40±0.04 | 0.58±0.03 |
| | SHD | 883.5±42.2 | 342.7±9.5 | 414.8±23.8 | 585.3±29.1 | 416.8±16.4 | 922.3±41.1 | 648.0±20.3 | 1382.5±42.3 | 1627.4±84.8 | 458.0±26.2 |
| | T (s) | 5.5±0.8 | 6.1±0.1 | 6.0±0.1 | 6.2±0.9 | 5.7±0.4 | 19.1±6.3 | 2.8±0.2 | 6.6±0.6 | 4806.5±2311.3 | 5.3±0.8 |
| P/fGES | $F1^-$ | 0.68±0.01 | 0.78±0.01 | 0.74±0.01 | 0.75±0.01 | 0.74±0.01 | 0.66±0.01 | 0.77±0.01 | 0.56±0.02 | 0.53±0.01 | 0.75±0.01 |
| | $F1^\rightarrow$ | 0.57±0.03 | 0.64±0.01 | 0.73±0.01 | 0.56±0.01 | 0.72±0.01 | 0.57±0.02 | 0.56±0.01 | 0.45±0.03 | 0.47±0.02 | 0.62±0.02 |
| | SHD | 1234.3±77.9 | 710.5±24.7 | 611.1±31.2 | 1055.5±39.4 | 617.5±27.1 | 1158.4±65.2 | 1051.6±31.0 | 1826.2±100.4 | 2442.4±115.6 | 871.6±32.0 |
| | T (s) | 9.3±1.1 | 9.0±0.4 | 8.1±0.4 | 9.0±1.3 | 7.9±0.4 | 342.6±283.8 | 4.4±0.2 | 11.7±1.7 | 570.3±1084.5 | 6.9±0.6 |
| PC-Stable | $F1^-$ | 0.72±0.01 | 0.73±0.01 | 0.54±0.01 | 0.82±0.01 | 0.54±0.01 | 0.67±0.01 | 0.80±0.01 | 0.61±0.01 | 0.45±0.36 | 0.68±0.01 |
| | $F1^\rightarrow$ | 0.50±0.01 | 0.37±0.01 | 0.18±0.00 | 0.56±0.01 | 0.25±0.00 | 0.38±0.01 | **0.61±0.01** | 0.29±0.23 | 0.44±0.01 | 0.40±0.01 |
| | SHD | 1256.9±26.9 | 1189.2±37.0 | 1771.5±42.8 | 792.2±36.3 | 1700.1±49.5 | 1520.4±42.6 | 724.8±24.7 | 2144.2±47.3 | 1718.2±41.7 | 1165.8±40.2 |
| | T (s) | 383.4±65.6 | 167.7±14.5 | 176.4±19.8 | 309.2±57.3 | 165.9±12.4 | 453.8±103.8 | 172.6±14.6 | 569.4±150.5 | 43121.0±35902.9 | 169.7±15.3 |
| fGES | $F1^-$ | 0.57±0.00 | 0.51±0.00 | 0.43±0.00 | 0.55±0.00 | 0.43±0.00 | 0.47±0.01 | 0.58±0.00 | 0.57±0.01 | 0.53±0.00 | 0.50±0.00 |
| | $F1^\rightarrow$ | 0.49±0.02 | 0.42±0.01 | 0.42±0.00 | 0.42±0.01 | 0.42±0.01 | 0.42±0.01 | 0.43±0.01 | 0.48±0.02 | 0.48±0.01 | 0.40±0.01 |
| | SHD | 2226.1±48.7 | 2301.3±17.7 | 2320.3±16.1 | 2337.9±29.4 | 2326.9±20.0 | 2512.1±37.0 | 2331.5±25.1 | 2259.6±49.2 | 3143.3±51.1 | 2396.8±22.3 |
| | T (s) | 769.1±104.8 | 823.2±26.6 | 787.0±19.9 | 907.6±35.2 | 813.5±28.0 | 907.6±44.5 | 832.4±27.6 | 646.5±28.6 | 1257.0±74.2 | 770.2±22.8 |
| Impro. ratio | $F1^-$ | 7.4% | 4.7% | 14.6% | 0.2% | 14.6% | 13.2% | -0.1% | 10.4% | 12.5% | 3.2% |
| | $F1^\rightarrow$ | 19.0% | 16.3% | 29.5% | 6.8% | 30.0% | 6.7% | -3.3% | 12.3% | 28.2% | 26.0% |
| | SHD | 29.2% | 63.5% | 89.6% | 0.7% | 88.9% | 38.1% | 25.2% | 17.9% | 22.4% | 32.3% |

**Baselines and Settings**  Six baselines are considered, including existing state-of-the-art methods and PEF-based methods. Specifically, given the rich literature on BN structure learning (Kitson et al., 2023), we collect 15 existing methods, including score-based, constraint-based, and hybrid methods, and conduct a preliminary testing of them (see Appendix C.2). The results indicate that fGES and PC-Stable are the only methods capable of maintaining F1 scores of around 0.5 within reasonable time when the variable number reaches 1000. Therefore, we choose these two as baselines. Besides, fGES and PC-Stable are also integrated into the estimation step of PEF, resulting in two new baselines: P/fGES and P/PC-Stable. Furthermore, to validate the effectiveness of Auto-SLE, we consider two alternative ensemble construction approaches: (i) default SLE, which contains the default fGES and PC-Stable, as well as variants with randomly chosen hyperparameter values for each of them; (ii) random SLE, which contains two variants with randomly chosen hyperparameter values for each of fGES and PC-Stable. Both of these SLEs consist of four member algorithms (same as our constructed SLE) and are integrated into PEF, yielding two baselines P/SLE(D) and P/SLE(R).

To prevent the compared methods from running prohibitively long, a runtime limit of 24 hours is set on each testing problem. All the experiments are conducted on a Linux server with an Intel Gold 6336Y CPU @ 2.40GHz, 96 cores, and 768GB of memory. Precise details of the experimental setup, including the preliminary testing results, SLE construction, benchmarks, metrics, and baselines, are in Appendix C. The codes for repeating our experiments are available in the supplementary.

## 4.2 RESULTS AND ANALYSIS

The partition step of PEF typically results in subproblems with 5%~10% variables of the original problem. For testing problems with 1000 and 10,000 variables, the subproblems have 50~100 and 500~1000 variables, respectively, which are problem sizes covered by training data. However, as the number of variables increases further (e.g., to 30,000), the subproblems will be much larger than the training problems. We first examine the performance on testing problems with 1000 and 10,000

Table 2: Results on problems with 10,000 variables. "-" means a solution is not found within the budget of 24 hours. The best performance in terms of accuracy is marked in **bold** and an underline.

| Problem ($\|V\|,\|E\|$) | | Alarm (10027, 13713) | Asia (10000, 11000) | Cancer (10000, 8800) | Child (10000, 13750) | Earthquake (10000, 8800) | Hailfinder (10024, 12996) | Healthcare (10003, 14148) | Mildew (10010, 14472) | Pigs (10143, 14978) | Survey (10002, 11003) |
|---|---|---|---|---|---|---|---|---|---|---|---|
| P/SLE | F1⁻ | **0.80** | **0.94** | **0.96** | **0.85** | **0.96** | **0.81** | **0.88** | **0.69** | **0.76** | **0.90** |
| | F1→ | **0.66** | **0.72** | **0.92** | **0.61** | **0.92** | **0.66** | **0.59** | **0.55** | **0.69** | **0.77** |
| | SHD | **6366** | **1717** | **705** | **6017** | **683** | **5400** | **5361** | **10911** | **7468** | **3334** |
| | T (s) | 95.5 | 35.6 | 24.4 | 80.8 | 24.2 | 828.6 | 50.0 | 360.3 | 2120.5 | 39.2 |
| P/SLE(D) | F1⁻ | 0.34 | 0.37 | 0.29 | 0.42 | 0.31 | 0.34 | 0.43 | 0.30 | 0.33 | 0.35 |
| | F1→ | 0.28 | 0.30 | 0.28 | 0.32 | 0.30 | 0.30 | 0.31 | 0.23 | 0.29 | 0.28 |
| | SHD | 42805 | 36831 | 40464 | 35660 | 37413 | 40151 | 37297 | 50139 | 41958 | 39476 |
| | T (s) | 4084.4 | 2754.9 | 3308.6 | 3383.0 | 2810.6 | 3865.9 | 3739.6 | 4532.6 | 7040.1 | 2931.4 |
| P/SLE(R) | F1⁻ | 0.49 | 0.59 | 0.42 | 0.69 | 0.43 | 0.45 | 0.71 | 0.38 | 0.02 | 0.55 |
| | F1→ | 0.25 | 0.25 | 0.15 | 0.41 | 0.14 | 0.28 | 0.28 | 0.19 | 0.01 | 0.19 |
| | SHD | 23673 | 18141 | 26652 | 12844 | 26065 | 25024 | 15483 | 32409 | 15551 | 21367 |
| | T (s) | 676.8 | 386.7 | 667.9 | 343.2 | 538.6 | 628.6 | 319.6 | 5563.9 | 7048.2 | 379.0 |
| P/PC-Stable | F1⁻ | 0.59 | 0.71 | 0.54 | 0.77 | 0.55 | 0.55 | 0.80 | 0.44 | 0.12 | 0.66 |
| | F1→ | 0.31 | 0.32 | 0.20 | 0.46 | 0.17 | 0.34 | 0.34 | 0.23 | 0.06 | 0.26 |
| | SHD | 16842 | 11262 | 17280 | 8932 | 16869 | 17125 | 10815 | 24847 | 16005 | 14046 |
| | T (s) | 210.1 | 145.0 | 147.3 | 145.2 | 133.3 | 219.0 | 156.8 | 508.7 | 7005.0 | 125.4 |
| P/fGES | F1⁻ | 0.34 | 0.37 | 0.29 | 0.42 | 0.31 | 0.34 | 0.43 | 0.30 | 0.33 | 0.35 |
| | F1→ | 0.28 | 0.30 | 0.28 | 0.32 | 0.30 | 0.30 | 0.31 | 0.23 | 0.29 | 0.28 |
| | SHD | 42805 | 36831 | 40464 | 35660 | 37413 | 40151 | 37297 | 50139 | 41958 | 39476 |
| | T (s) | 4037.6 | 2796.0 | 3331.4 | 3369.7 | 2833.4 | 3669.7 | 3755.3 | 4439.0 | 4179.0 | 2905.7 |
| PC-Stable | F1⁻ | - | - | - | - | - | - | - | - | - | - |
| | F1→ | - | - | - | - | - | - | - | - | - | - |
| | SHD | - | - | - | - | - | - | - | - | - | - |
| | T (s) | 86400.0 | 86400.0 | 86400.0 | 86400.0 | 86400.0 | 86400.0 | 86400.0 | 86400.0 | 86400.0 | 86400.0 |
| fGES | F1⁻ | - | - | - | - | - | - | - | - | - | - |
| | F1→ | - | - | - | - | - | - | - | - | - | - |
| | SHD | - | - | - | - | - | - | - | - | - | - |
| | T (s) | 86400.0 | 86400.0 | 86400.0 | 86400.0 | 86400.0 | 86400.0 | 86400.0 | 86400.0 | 86400.0 | 86400.0 |
| Impro. ratio | F1⁻ | 36.0% | 31.4% | 78.8% | 10.8% | 75.7% | 47.6% | 10.4% | 54.5% | 133.0% | 36.0% |
| | F1→ | 115.6% | 127.2% | 225.4% | 33.9% | 208.0% | 94.7% | 70.9% | 135.0% | 135.8% | 180.0% |
| | SHD | 62.2% | 84.8% | 95.9% | 32.6% | 96.0% | 68.5% | 50.4% | 56.1% | 52.0% | 76.3% |

variables to answer RQ1. Specifically, based on each network selected from *bnlearn*, we generate 10 testing problems (with different random seeds), test each method on these problems, and report the mean ± std in terms of the evaluation metrics as well as stastical test results in Table 1. For the performance evaluation involving 10,000 variables, due to the very long runtime of the baselines, we generate one test problem based on each network and report the testing results in Table 2.

The first observation from Table 1 is that P/SLE consistently achieves significantly higher accuracy across all three metrics compared to the baselines, except for the F1 metrics on Healthcare testing problems, where P/SLE performs slightly worse than PC-Stable. Table 2 shows that the superiority of P/SLE becomes more pronounced on testing problems with 10,000 variables. In call cases it achieves substantially higher accuracy than all baselines across all accuracy metrics. Notably, compared to the best performance achieved by the baselines, P/SLE often achieves improvements in F1 Adjacent of over 30% and up to 133%, and improvements in F1 Arrowhead and SHD of over 50% and even up to 225%. Since the difference between P/SLE and P/PC-Stable (P/fGES) lies in the use of a SLE in the estimation step instead of a single algorithm, the consistent advantages of P/SLE over them confirm that using SLEs can stably achieve high learning accuracy across sub-problems. On the other hand, we also observe that P/SLE(D) and P/fGES obtain identical learning accuracy. This is because, in the default ensemble, the output of fGES always has the best BIC score among all member algorithms, thus making it consistently being chosen as the final output. While P/SLE(R), with a randomly constructed SLE, can avoid this issue, it fails to achieve satisfactory learning accuracy, which in some cases is even worse than P/PC-Stable and P/fGES that do not use SLEs. Therefore, the advantages of P/SLE over P/SLE(D) and P/SLE(R) highlight the effectiveness of Auto-SLE in producing high-quality SLEs with complementary member algorithms.

The second observation is that on testing problems with 1000 variables, P/PC-Stable and P/fGES often achieve higher learning accuracy than PC-Stable and fGES, respectively. Moreover, when the number of variables reaches 10,000, PC-Stable and fGES are unable to find solutions within 24 hours, while P/PC-Stable and P/fGES are able to. These findings show that PEF framework can indeed enhance the capabilities of handling large BNs, which is consistent with the observations in (Gu & Zhou, 2020). Finally, all PEF-based methods generally consume much less runtime than non-PEF-based methods, attributed to the underlying divide-and-conquer strategy. Among PEF-based methods, P/SLE often has the shortest or close to the shortest runtime, and for most testing problems, it consistently outputs the final solution within a reasonable time (less than 1000 seconds). In summary, all the above findings affirmatively answer RQ1, i.e., the SLE constructed by Auto-SLE substantially improves PEF in learning the structure of large BNs.

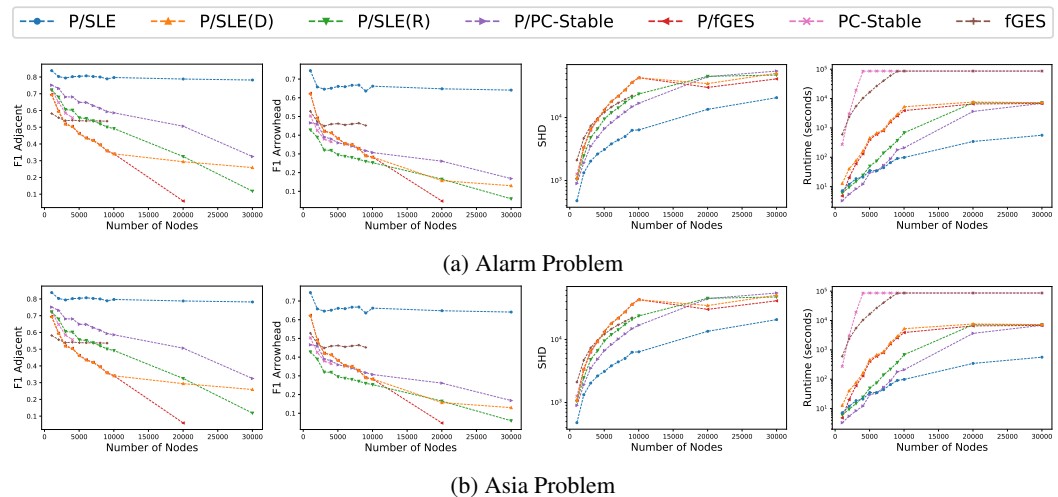

(a) Alarm Problem

(b) Asia Problem

Figure 2: Performance curves on Alarm and Asia problems with up to 30,000 variables. SHD and runtime are plotted on log scale.

Table 3: Testing results on Yeast and WS problems. "P/SLE-b" represents the best performance achieved among P/SLE(R), P/SLE(D), P/fGES, and P/PC-Stable. The best performance in terms of accuracy metrics is marked with an underline, and the performance that is not significantly different from the best performance is indicated in **bold**.

| Problem ($|V|$, $|E|$) | | WS (1000, 2000) | WS (10000, 20000) | Yeast (4441, 12873) |
|---|---|---|---|---|
| P/SLE | $F1^-$ | **0.80±0.01** | **0.79** | **0.106** |
| | $F1^\to$ | **0.51±0.01** | **0.49** | **0.084** |
| | SHD | **1240.2±47.7** | **12611** | **20912** |
| | T (s) | 9.8±0.3 | 83.2 | 7105.4 |
| P/SLE-b | $F1^-$ | 0.79±0.01 | 0.72 | 0.003 |
| | $F1^\to$ | **0.51±0.01** | 0.40 | 0.001 |
| | SHD | 1252.4±32.2 | 15760 | 33162 |
| | T (s) | 3.7±0.3 | 363.1 | 5771.32 |
| PC-Stable | $F1^-$ | 0.75±0.01 | - | - |
| | $F1^\to$ | 0.42±0.01 | - | - |
| | SHD | 1437.4±21.2 | - | - |
| | T (s) | 176.2±6.5 | 86400.0 | 86400.0 |
| fGES | $F1^-$ | 0.67±0.00 | - | 0.066 |
| | $F1^\to$ | 0.43±0.01 | - | 0.056 |
| | SHD | 2401.0±22.3 | - | 29847 |
| | T (s) | 1433.3±61.4 | 86400.0 | 178257.4 |
| Impro. ratio | $F1^-$ | 2.2% | 10.6% | 60.6% |
| | $F1^\to$ | -0.6% | 21.1% | 50.0% |
| | SHD | 1.0% | 20.0% | 29.9% |

## 4.3 GENERALIZATION TO LARGER PROBLEMS

We now investigate RQ2. Specifically, we generate Alarm and Asia testing problems with 20,000 and 30,000 variables, and plot in Figure 2 the performance of the compared methods as the variable number ranges from 1000 to 30,000 (detailed results can be found in Appendix C.5). Note that when the variable number exceeds 20,000, the subproblems resulted from the partition step of PEF would be much larger in size than the training problems. It can be seen from Figure 2 that P/SLE generalizes well to larger problems, maintaining relatively stable learning accuracy. In contrast, the performance of all baselines deteriorates rapidly as the number of variables increases.

## 4.4 GENERALIZATION TO PROBLEMS WITH NO BLOCK STRUCTURE

The above results have demonstrated that P/SLE can stably achieve high learning accuracy for networks with a block structure. Of course, there are many real-world networks without any block

structure (Olesen & Madsen, 2002). Although PEF-based methods are not specifically designed for such networks, it is worth testing P/SLE on them to provide a complete spectrum of its performance. Specifically, we apply P/SLE for gene expression data analysis, a traditional application of structure leanring for BN. We use the largest publicly available gene dataset Yeast (Schaffter et al., 2011) involving 4,441 nodes and 12,873 edges, where the underlying networks are commonly referred to as gene regulatory networks. Besides, we generate small-world networks with 1000 and 10,000 nodes, using *igraph* (Csardi et al., 2006) based on the Watts-Strogatz (WS) model (Watts & Strogatz, 1998). We choose the WS model because: (i) it has no block structure, and (ii) it is not included in the *bnlearn* repository, thereby enabling evaluation of the generalization of P/SLE to network characteristics beyond the training set. As before, we generate 10 testing problems based on the network with 1000 nodes and one testing problem based on the network with 10,000 nodes. The testing results are presented in Table 3.

One can observe that P/SLE still achieves competitive learning accuracy on these networks. On WS problems with 1000 variables, it always achieves the best performance or the performance not significantly different from the best, across all accuracy metrics. On WS problems with 10,000 variables and Yeast, the advantages of P/SLE become pronounced, similar to the previous observations on networks with block structures. In summary, these findings show the generalization ability of the constructed SLE across network characteristics beyond the training problem set.

## 5 CONCLUSION

In this work, we introduced the idea of using SLEs for BN structure learning and proposed Auto-SLE, an automatic approach that can largely reduce human efforts in building high-quality SLEs. Extensive experiments showed that our method P/SLE could consistently achieve high accuracy in learning large BNs and generalize well across problem sizes and network characteristics.

**Limitations** There are two main limitations of this work. First, while using an SLE by running its member algorithms in parallel would not significantly increase the wall-clock runtime compared to running a single algorithm, executing them sequentially in the absence of multi-core compute leads to significantly longer runtime. A potential solution is to train a selection model that predicts the best-performing algorithm in the SLE for a given problem, and runs that algorithm only. Second, the generalization ability of P/SLE relies on a training set with diverse networks of varying sizes. If such a set cannot be collected in practice, then it may not generalize well. Moreover, as aforementioned, PEF-based methods are most suitable for learning BNs with a block structure to some extent. For BNs with no block structure at all, P/SLE may not be the best choice, and in these cases the SLE is more suitable as a standalone learning method rather than being integrated into the PEF framework.

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

## A   THE PARTITION-ESTIMATION-FUSION (PEF) FRAMEWORK

This section presents the implementation details of the partition-estimation-fusion (PEF) Gu & Zhou (2020) framework. PEF consists of the following three steps.

- **Partition**: The nodes are divided into clusters using a modified hierarchical clustering (MHC) algorithm.
- **Estimation**: An existing structure learning method is applied to estimate a subgraph on each cluster of nodes.

---

**Algorithm 2:** Modified Hierarchical Clustering

---

Hierarchical clustering given the dissimilarity matrix $D = (d(i,j))_{d \times d}$;
Generate the dendrogram $T_D$ of the hierarchical clustering;
Choose $p$ by Eq. (9) and $l$ by Eq. (10);
Relabel clusters in $C \leftarrow C_l$ so that $S_1 \leq \cdots \leq S_{d-l}$;
**while** $|C| > p$ **do**
$\quad | \quad (i^*, j^*) \leftarrow \arg\min_{(i,j)}\{d(C_i, C_j) : i < j \land j > p\}$;
$\quad | \quad C_{i^*} \leftarrow C_{i^*} \cup C_{j^*}, C \leftarrow C \backslash \{C_{j^*}\}$;
**end**
**return** $C = \{C_1, C_2, \ldots, C_p\}$;

---

- **Fusion**: Merge estimated subgraphs into one DAG containing all the nodes.

Suppose we have observed $m$ iid observations of random variables $X_1, \ldots, X_d$, denoted as $D \in \mathbb{R}^{m \times d}$. Denote the $i$-th column of $D$ as $\mathbf{x}_i$. Given $D$, the goal is to learn a DAG structure $\mathcal{G} = (V, E)$ that accurately reflects the conditional dependencies among $X_1, \ldots, X_d$.

## A.1 PARTITION

The partition step (P-step) of the PEF involves partitioning nodes into clusters. From this procedure, $p$ clusters, denoted as $C_i$ for $i = 1, 2, \ldots, p$, are generated. This utilizes a modified hierarchical clustering (MHC) approach, equipped with average linkage, that autonomously determines the number of clusters $p$. The distance between two nodes $i$ and $j$ in PEF is defined by a specific equation,

$$d(i,j) = 1 - |r_{ij}| \in [0, 1] \tag{7}$$

where $r_{ij} = cor(X_i, X_j)$ represents the correlation between $X_i$ and $X_j$ for $i, j = 1, 2, \ldots, d$. The correlation is calculated using covariance $cov(\mathbf{x}_i, \mathbf{x}_j)$ and standard deviations $\sigma_{\mathbf{x}_i}, \sigma_{\mathbf{x}_j}$, and the following equation.

$$cor(X_i, X_j) = \frac{cov(\mathbf{x}_i, \mathbf{x}_j)}{\sigma_{\mathbf{x}_i} \sigma_{\mathbf{x}_j}} \tag{8}$$

PEF mandates that the minimum cluster size should be $0.05d$. For each $h = 0, 1, ..., d-1$, $C_h$ designates the clusters formed during the $h-$th iteration of bottom-up hierarchical clustering. Specifically, $C_0 = \{\{1\}, \{2\}, \ldots, \{d\}\}$ consists of $p$ singleton clusters, while $C_{p-1} = \{\{1, 2, \ldots, d\}\}$ denotes a single cluster encompassing all $d$ nodes. Let $p_i$ signify the count of big clusters in $C_i$. PEF selects a particular $p$ according to the following equation:

$$p = \min\{p_{max}, \max_{0 \leq i \leq d-1} p_i\} \tag{9}$$

where $p_{max} \leq 20$ represents a user-defined maximum count of big clusters.

Let $l$ represent the topmost level on the dendrogram housing $p$ big clusters, expressed as the following equation:

$$l = \arg\max_{0 \leq i \leq d-1}\{i : p_i = p\} \tag{10}$$

Clusters in $C_l$ are then relabeled in descending order based on their size, so $S_1 \geq S_2 \geq \cdots \geq S_{d-l}$, with $S_i = |C_i|$. The first $p$ clusters are subsequently identified as the primary big clusters of interest. PEF proceeds to allocate the leftover small clusters to these $p$ big clusters, which is accomplished by repetitively merging the two nearest clusters, provided one is a small cluster. The pseudocode of the MHC algorithm is detailed in Algorithm 2, where the dissimilarity matrix $D = (d(i,j))_{d \times d}$ is calculated by Eq. (7). In the experiments, we limit the size of the cluster to be less than 10% of the original problem to prevent the occurrence of excessively large subproblems.

## A.2 ESTIMATION

During the estimation step (E-step), the PEF determines the structure of each subgraph individually. Within the PEF framework, this step acts like a blackbox, allowing users to employ any structure

learning algorithm to estimate subgraphs without needing in-depth knowledge of its technical details. Typically, this step yields $p$ partial DAGs (PDAGs). It is noteworthy that both DAGs and complete PDAGs (CPDAGs) are subsets of PDAGs. If the time complexity of a structure learning technique surpasses $O(d^2)$, the time required to learn small subgraphs during the E-step becomes considerably less than that needed to estimate an entire DAG. Assuming that during the partition step nodes were divided into $p$ clusters $C_1, C_2, \ldots, C_p$ and the duration to learn a PDAG on $C_i$ is $t_i$, parallelizing the learning of $p$ subgraphs across $p$ cores can reduce the E-step duration to $\max\{t_i : i = 1, 2, \ldots, p\}$.

### A.3 Fusion

In the fusion step (F-step), a hybrid methodology is employed to learn the full DAG structure from the estimated subgraphs (obtained in the E-step). This step unfolds in two stages. First, PEF generates a candidate edge set $A$ to restrict the search space. Through a series of statistical tests, PEF discerns a subset, $A^*$, comprising candidate edges between subgraphs. Consequently, the candidate edge set $A$ consists of $A^*$ and all edges learned in each subgraph from the E-step. Second, PEF optimize the DAG structure by iteratively updating edges within set $A$ based on a modified BIC score. The final output of the F-step is a DAG.

## B Proofs

### B.1 Proof of Fact 1

**Fact 2.** $Q(\cdot)$ *is a monotone submodular function, i.e., for any two SLEs* $\mathbb{A}, \mathbb{A}' \subset \Theta$ *and any* $\theta \in \Theta$, *it holds that* $Q(\mathbb{A}) \leq Q(\mathbb{A} \cup \mathbb{A}')$ *and* $Q(\mathbb{A} \cup \mathbb{A}' \cup \{\theta\}) - Q(\mathbb{A} \cup \mathbb{A}') \leq Q(\mathbb{A} \cup \{\theta\}) - Q(\mathbb{A})$.

*Proof.* By definition, $Q(\mathbb{A}, D) = \max_{\theta \in \mathbb{A}} Q(\theta, D)$, then it holds that $Q(\mathbb{A} \cup \mathbb{A}', D) = \max_{\theta \in \mathbb{A} \cup \mathbb{A}'} Q(\theta, D) \geq \max_{\theta \in \mathbb{A}}, Q(\theta, D)$. The monotonicity holds.

To prove submodularity, we have

$$
\begin{aligned}
Q(\mathbb{A} \cup \{\theta\}) - Q(\mathbb{A}) &= \frac{1}{|T|} \sum_{D \in T} [Q(\mathbb{A} \cup \{\theta\}, D) - Q(\mathbb{A}, D)] \quad \text{by definition of } Q(\cdot) \\
&= \frac{1}{|T|} \sum_{D \in T} [Q(\theta, D) - Q(\mathbb{A}, D)]^+ \\
&\geq \frac{1}{|T|} \sum_{D \in T} [Q(\theta, D) - Q(\mathbb{A} \cup \mathbb{A}', D)]^+ \quad \text{by monotonicity} \\
&= Q(\mathbb{A} \cup \mathbb{A}' \cup \{\theta\}) - Q(\mathbb{A} \cup \mathbb{A}').
\end{aligned}
\tag{11}
$$

The proof is complete. $\qquad\square$

### B.2 Proof of Theorem 1

**Theorem 2.** *Using a hyperparameter optimization procedure that, in each iteration of Auto-SLE, returns* $\hat{\theta}$ *within* $\epsilon$-*absolute error of the maximum of* $\Delta(\theta|\mathbb{A})$, *i.e.,* $\Delta(\hat{\theta}|\mathbb{A}) \geq \Delta(\theta^\circ|\mathbb{A}) - \epsilon$, *then the quality* $Q(\mathbb{A})$ *of the SLE constructed by Auto-SLE is bounded by*

$$
Q(\mathbb{A}) \geq (1 - 1/e) \cdot Q(\mathbb{A}^*) - k\epsilon,
\tag{12}
$$

*where* $\mathbb{A}^*$ *is the optimal SLE to the SLE construction problem in Definition 1. Alternatively, if* $\hat{\theta}$ *is within* $\epsilon$-*relative error of* $\Delta(\theta^\circ|\mathbb{A})$, *i.e.,* $\Delta(\hat{\theta}|\mathbb{A}) \geq \Delta(\theta^\circ|\mathbb{A}) \cdot (1 - \epsilon)$, *then the quality* $Q(\mathbb{A})$ *of the SLE constructed by Auto-SLE is bounded by*

$$
Q(\mathbb{A}) \geq (1 - 1/e^{1-\epsilon}) \cdot Q(\mathbb{A}^*).
\tag{13}
$$

*Proof.* Order the candidate algorithms in $\mathbb{A}^*$ as $\{\theta_1^* \ldots \theta_k^*\}$. We denote $\mathbb{A} = \{\theta_1, \theta_2, \ldots, \theta_k\}$ where $\theta_i$ is the algorithm added to $\mathbb{A}$ in the $i$-th iteration of Auto-SLE. Let $\mathbb{A}_i = \{\theta_1, \ldots, \theta_i\}$ and

let $\Delta(\theta|\mathbb{A}) = Q(\mathbb{A} \cup \{\theta\}) - Q(\mathbb{A})$ denote the performance improvement brought by adding $\theta$ to $\mathbb{A}$.

In the first case where $\Delta(\hat{\theta}|\mathbb{A}) \geq \Delta(\theta^\circ|\mathbb{A}) - \epsilon$, for all positive integers $i < l \leq k$, we have:

$$
\begin{aligned}
Q(\mathbb{A}^*) &\leq Q(\mathbb{A}^* \cup \mathbb{A}_i) && \text{by monotonicity} \\
&= Q(\mathbb{A}_i) + \sum_{j=1}^{k} \Delta(\theta_j^*|\mathbb{A}_i \cup \{\theta_1^*, \ldots, \theta_{j-1}^*\}) && \text{by telescoping sum} \\
&\leq Q(\mathbb{A}_i) + \sum_{\theta \in \mathbb{A}^*} \Delta(\theta|\mathbb{A}_i) && \text{by submodularity} \\
&\leq Q(\mathbb{A}_i) + \sum_{\theta \in \mathbb{A}^*} \Delta(\theta^\circ|\mathbb{A}_i) && \text{by definition of } \theta^\circ \\
&\leq Q(\mathbb{A}_i) + \sum_{\theta \in \mathbb{A}^*} (Q(\mathbb{A}_{i+1}) - Q(\mathbb{A}_i) + \epsilon) && \text{by } \Delta(\theta^\circ|\mathbb{A}) \leq \Delta(\hat{\theta}|\mathbb{A}) + \epsilon \\
&\leq Q(\mathbb{A}_i) + k(Q(\mathbb{A}_{i+1}) - Q(\mathbb{A}_i) + \epsilon).
\end{aligned}
\tag{14}
$$

Let $\delta_i = Q(\mathbb{A}^*) - Q(\mathbb{A}_i)$, which allows us to rewrite the above equation as $\delta_i \leq k(\delta_i - \delta_{i+1} + \epsilon)$, then $\delta_{i+1} \leq (1 - \frac{1}{k})\delta_i + \epsilon$. Hence, we have

$$
\begin{aligned}
\delta_l &\leq (1 - \frac{1}{k})^l \delta_0 + k\epsilon \cdot [1 - (1 - \frac{1}{k})^l] \\
&\leq e^{-l/k}\delta_0 + k\epsilon && \text{by } 1 - x \leq e^{-x} \text{ for all } x \in \mathbb{R} \\
&= e^{-l/k}(Q(\mathbb{A}^*) - Q(\mathbb{A}_0)) + k\epsilon \\
&= e^{-l/k}Q(\mathbb{A}^*) + k\epsilon && \text{by that } \mathbb{A}_0 = \varnothing \text{ and } Q(\mathbb{A}_0) = 0
\end{aligned}
\tag{15}
$$

Rearranging $\delta_l = Q(\mathbb{A}^*) - Q(\mathbb{A}_l) \leq e^{-l/k}Q(\mathbb{A}^*) + k\epsilon$, we have

$$
Q(\mathbb{A}_l) \geq (1 - e^{-l/k}) \cdot Q(\mathbb{A}^*) - k\epsilon.
\tag{16}
$$

Since the SLE found by Auto-SLE is $\mathbb{A}_k$, then we have

$$
Q(\mathbb{A}_k) \geq (1 - 1/e) \cdot Q(\mathbb{A}^*) - k\epsilon.
\tag{17}
$$

In the second case where $\Delta(\hat{\theta}|\mathbb{A}) \geq \Delta(\theta^\circ|\mathbb{A}) \cdot (1 - \epsilon)$. Similarly, for all positive integers $i < l \leq k$, we have:

$$
\begin{aligned}
Q(\mathbb{A}^*) &\leq Q(\mathbb{A}_i) + \sum_{\theta \in \mathbb{A}^*} (Q(\mathbb{A}_{i+1}) - Q(\mathbb{A}_i))/(1 - \epsilon) && \text{by } \Delta(\theta^\circ|\mathbb{A}) \leq \Delta(\hat{\theta}|\mathbb{A})/(1 - \epsilon) \\
&= Q(\mathbb{A}_i) + \frac{k}{1 - \epsilon}(Q(\mathbb{A}_{i+1}) - Q(\mathbb{A}_i)).
\end{aligned}
\tag{18}
$$

Similarly, let $\delta_i = Q(\mathbb{A}^*) - Q(\mathbb{A}_i)$ and use the above procedure, we have

$$
Q(\mathbb{A}_l) \geq (1 - e^{-l(1-\epsilon)/k}) \cdot Q(\mathbb{A}^*).
\tag{19}
$$

Let $l = k$, we have

$$
Q(\mathbb{A}_k) \geq (1 - 1/e^{1-\epsilon}) \cdot Q(\mathbb{A}^*).
\tag{20}
$$

The proof is complete. $\qquad\square$

## C    DETAILS OF THE EXPERIMENTS

Throughout the experiments, we set a random seed as 1024, ensuring the reproducibility of our experiments. The codes for repeating our experiments can be found in the supplementary.

### C.1 EVALUATION METRICS

Let M1 be the true MEC of the DAG and M2 be the estimated MEC, the F1 score for adjacencies (F1 Adjacent) is calculated as the harmonic mean of precision and recall: 2TP/(2TP+FP+FN), where TP is the number of adjacencies shared by M1 and M2, FP is the number of adjacencies in M2 but not in M1, and FN is the number of adjacencies in M1 but not in M2. The F1 score for arrowhead (F1 Arrowhead) is calculated in a similar way. An arrowhead is taken to be in M1 and M2 for each variable A and B such that A → B in both M1 and M2, and an arrowhead is taken to be in one MEC but not the other when for each variable A and B such that A → B in one but A ← B in the other, or A–B (no directions) in the other, or A and B are not adjacent in the other.

The structural Hamming distance (SHD) is defined as the number of edge insertions, deletions or flips in order to transform the learned DAG to the ground truth.

### C.2 PRELIMINARY TESTING

We collect 15 existing methods, listed below.

- Score-based combinatorial search methods: HC Chickering et al. (2004), TABU Bouckaert (1995), CCDr Aragam & Zhou (2015), fGES Ramsey et al. (2017)
- Score-based continuous optimization methods: NOTEARS Zheng et al. (2018), GOLEM Zhu et al. (2021)
- Constraint-based methods: PC-Stable Colombo et al. (2014), GS Margaritis et al. (2003), IAMB Tsamardinos et al. (2003b), Fast-IAMB Tsamardinos et al. (2003b), IAMB-FDR Pena (2008), Inter-IAMB Yaramakala & Margaritis (2005)
- Hybrid methods: MMHC Tsamardinos et al. (2006), RSMAX2 Friedman et al. (2013), H2PC Gasse et al. (2014)

We collect the open-source implementations of these methods. Most of the implementations are collected from the *bnlearn* Scutari (2010) repository [1]; CCDr [2], NOTEARS [3], GOLEM [4] are collected from Github; fGES and PC-Stable are collected from *TETRAD* Ramsey et al. (2018) [5].

We generate testing problems based on two random graph models, Erdös-Rényi (ER) Erdős et al. (1960) and scale-free (SF) Barabási & Bonabeau (2003), where the edge number is set to be two times the node number. Specifically, each testing problem involves 1000 variables, has Gaussian noise, and the observation number $m = 1000$. Based on each graph model, 10 different testing problems are generated, resulting in a total of 20 testing problems. Each method is then applied to these testing problems, with a runtime limit of 3600 seconds for each problem. Table 4 presents the average performance of these methods on all 20 testing problems, measured by F1 score (Adjacent) and runtime. The results indicate that fGES and PC-Stable are the top-performing methods, consistently maintaining F1 scores above 0.5 when the variable count reaches 1000. While HC and TABU also achieve F1 scores above 0.5, their runtime is significantly longer, making them impractical for testing problems involving 10,000 variables. As a result, they are excluded from the comparison experiments.

### C.3 AUTOMATIC CONSTRUCTION OF THE SLE

**Algorithm Configuration Space** During the SLE construction process, the algorithm configuration space is defined by two candidate algorithms and their hyperparameters.

- PC-Stable Colombo et al. (2014) with two hyperparameters: significance threshold of CI tests within the interval $\alpha \in [0.01, 0.2]$ and the search's maximum depth interval $m \in [1, 1000]$

---

[1] `https://www.bnlearn.com/bnrepository` (Creative Commons Attribution-Share Alike License).

[2] `https://github.com/itsrainingdata/ccdrAlgorithm` (license not specified)

[3] `https://github.com/xunzheng/notears` (Apache-2.0 license)

[4] `https://github.com/ignavierng/golem` (Apache-2.0 license)

[5] `https://www.ccd.pitt.edu/tools` (GNU General Public License (GPL) v2 license)

Table 4: Preliminary testing results. "-" means not returning solutions within a time budget of 3600s.

| Method | F1 | runtime (s) |
|---|---|---|
| PC-Stable | 0.71 | 576.21 |
| fGES | 0.66 | 625.86 |
| HC | 0.528 | 3170.22 |
| TABU | 0.528 | 3181.777 |
| CCDr | 0.392 | 834.781 |
| MMHC | 0.331 | 1210.805 |
| RSMAX2 | 0.326 | 1416.733 |
| GOLEM | 0.32 | 3622.433 |
| IAMB-FDR | 0.271 | 2021.871 |
| Inter-IAMB | 0.189 | 2865.472 |
| IAMB | 0.157 | 2943.536 |
| Fast-IAMB | 0.107 | 3351.702 |
| NOTEARS | 0.099 | 3604.744 |
| GS | - | 3600.32 |
| H2PC | - | 3610.257 |

- fGES Ramsey et al. (2017) with three hyperparameters: structural penalty of the BIC score within interval $\lambda \in [1.0, 1000.0]$, the maximum number of parents for a single node during the search process within interval $m \in [1, 1000]$, and the option to use the faithfulness assumption or not.

We use the implementations of them from the causal discovery tool box *TETRAD* Ramsey et al. (2018) [6]. SMAC (version 3) Lindauer et al. (2022) [7] is used as the hyperparameter optimization procedure.

**Training Problem Set**  For Auto-SLE, we used the data generation method and randomly produced 100 training problem instances. In order to ensure that the training data is diverse, comprehensive, and representative, each instance was derived by the following steps: (i) selecting a base network at random from the *bnlearn* Scutari (2010) repository [8]; (ii) replicating it a random number of times, while making sure the total node count is less than or equal to 1000, and connecting the replicas by adding 10% of edges between them randomly, ensuring the final network remains a DAG; (iii) utilizing the complete DAG and $X_i = \phi\left(X_{\mathrm{pa}(i)}\right) + \varepsilon_i$, to produce 1,000 observational data records. Here the weights in the linear function $\phi$ and the standard deviations of Gaussian noises are sampled uniformly from $[-1, -0.5] \cup [0.5, 1]$ and $[0, 1]$, respectively; (iv) re-scaling the observation data so that all data columns possess the same mean and standard deviation. These 100 training problem instances are independent of testing data and are exclusively for the construction of the SLE.

**The Constructed SLE**  The constructed SLE contains four member algorithms, as running more iterations of Auto-SLE brings negligible improvement (smaller than 0.1) in SLE's performance on the training set. Specifically, the training performance ()in terms of the sum of F1 Adjacent and F1 Arrowhead) progress is: ite 1 (143.4) → ite 2 (156.1) → ite 3 (158.1) → ite 4 (158.5) → ite 5 (158.5). The SLE is detailed in Table 5. It is interesting to find that the SLE only contains fGES, meaning PC-Stable has not defeated fGES in the construction process.

---

[6]https://www.ccd.pitt.edu/tools (GNU General Public License (GPL) v2 license)

[7]https://automl.github.io/SMAC3/main (3-clause BSD license)

[8]https://www.bnlearn.com/bnrepository (Creative Commons Attribution-Share Alike License)

Table 5: The SLE contructed by Auto-SLE.

| Member Algorithm | Candidate Algorithm | Hyperparameter Values |
|---|---|---|
| 1 | fGES | $\lambda = 5.872727477498224$, $m = 185$, and without faithfulness assumption |
| 2 | fGES | $\lambda = 20.60452402709974$, $m = 17$, and without faithfulness assumption |
| 3 | fGES | $\lambda = 2.537674798471765$, $m = 91$, and without faithfulness assumption |
| 4 | fGES | $\lambda = 5.624595350676138$, $m = 11$, and without faithfulness assumption |

## C.4 DEFAULT SLE AND RANDOM SLE

Default SLE and random SLEs are constructed in alternative ways other than Auto-SLE. The former contains the default fGES and PC-Stable, as well as variants with randomly chosen hyperparameter values for each of them; The latter contains two variants with randomly chosen hyperparameter values for each of fGES and PC-Stable. According to the causal discovery toolbox *TETRAD* Ramsey et al. (2018), the default hyperparameter values for PC-Stable is: $\alpha = 0.05$ and $m = 1000$; the default hyperparameter values for fGES is $\lambda = 1.0$, $m = 1000$, and without faithfulness assumption. Finally, SLE (Default) and SLE (Random) are shown in Table 6 and Table 7.

Table 6: Default SLE

| Member Algorithm | Candidate Algorithm | Hyperparameter Values |
|---|---|---|
| 1 | PC-Stable | $\alpha = 0.05$, and $m = 1000$ |
| 2 | fGES | $\lambda = 1.0$, $m = 1000$, and without faithfulness assumption |
| 3 | PC-Stable | $\alpha = 0.08399128452994686$, and $m = 850$ |
| 4 | fGES | $\lambda = 797.254519880674$, $m = 871$, and without faithfulness assumption |

Table 7: Random SLE.

| Member Algorithm | Candidate Algorithm | Hyperparameter Values |
|---|---|---|
| 1 | PC-Stable | $\alpha = 0.08399128452994686$, and $m = 850$ |
| 2 | fGES | $\lambda = 797.254519880674$, $m = 871$, and without faithfulness assumption |
| 3 | PC-Stable | $\alpha = 0.10744707122087944$, and $m = 980$ |
| 4 | fGES | $\lambda = 792.8350933385117$, $m = 456$, and with faithfulness assumption |

## C.5 TESTING RESULTS ON LARGER ALARM PROBLEMS AND ASIA PROBLEMS

Table 8 and Table 9 present the testing results of the compared methods on the Alarm and Asia problems with up to 30,000 variables, respectively.

## C.6 COMPUTE RESOURCE

Unless otherwise indicated, all experiments in this work are conducted on a Linux server equipped with an Intel(R) Xeon(R) Gold 6336Y CPU @ 2.40GHz, 96 cores, and 768GB of main memory. The system version is Ubuntu 22.04.2 LTS.

Table 8: Testing results on Alarm problems with up to 30000 variables, in terms of F1 Adjacent, F1 Arrowhead, SHD, and runtime. On each network, the mean ± std performance obtained by each method on 10 problems is reported. The best performance in terms of accuracy metrics is marked with an underline, and the performance that is not significantly different from the best performance (according to a Wilcoxon signed-rank test with significance level $p = 0.05$) is indicated in **bold**.

| Method | Problem ($|V|$, $|E|$) | Alarm-1000 (1036, 1417) | Alarm-2000 (2035, 2783) | Alarm-3000 (3034, 4150) | Alarm-4000 (4033, 5516) | Alarm-5000 (5032, 6882) | Alarm-6000 (6031, 8248) | Alarm-7000 (7030, 9614) | Alarm-8000 (8029, 10981) | Alarm-9000 (9028, 12347) | Alarm-10000 (10027, 13713) | Alarm-20000 (20017, 27375) | Alarm-30000 (30007, 41037) |
|---|---|---|---|---|---|---|---|---|---|---|---|---|---|
| P/SLE | F1 Adjacent | **83.8** | **80.3** | **79.4** | **80.2** | **80.4** | **80.7** | **80.3** | **80.0** | **78.9** | **79.7** | **78.8** | **78.2** |
|  | F1 Arrowhead | **74.5** | **65.8** | **64.6** | **65.1** | **66.1** | **65.9** | **66.7** | **66.8** | **63.6** | **66.2** | **64.8** | **64.1** |
|  | SHD | **478.0** | **1326.0** | **2030.0** | **2638.0** | **3117.0** | **3817.0** | **4409.0** | **4999.0** | **6272.0** | **6366.0** | **13491.0** | **20658.0** |
|  | Runtime (s) | 7.2 | 12.0 | 18.6 | 21.1 | 35.2 | 34.9 | 44.0 | 65.7 | 91.0 | 98.6 | 345.6 | 563.6 |
| P/SLE(D) | F1 Adjacent | 69.5 | 59.5 | 51.9 | 50.4 | 46.2 | 43.6 | 42.1 | 39.5 | 35.9 | 34.1 | 29.4 | 25.9 |
|  | F1 Arrowhead | 62.3 | 49.2 | 42.1 | 41.3 | 38.2 | 35.5 | 35 | 32.9 | 29 | 28.3 | 15.7 | 13 |
|  | SHD | 1086 | 3361 | 6549 | 9357 | 13371 | 18043 | 21892 | 27495 | 36108 | 42805 | 34557 | 50319 |
|  | Runtime (s) | 12.9 | 40.6 | 76.5 | 162.3 | 455.9 | 670.4 | 868.5 | 1755 | 2874.8 | 5262.8 | 7603.4 | 7263 |
| P/SLE(R) | F1 Adjacent | 72.2 | 68.0 | 60.5 | 60.2 | 55.6 | 55.2 | 53.8 | 52.0 | 50.1 | 49.2 | 32.5 | 11.8 |
|  | F1 Arrowhead | 42.8 | 38.9 | 32.0 | 31.9 | 29.5 | 28.7 | 28.1 | 27.1 | 26.0 | 25.3 | 16.5 | 6.0 |
|  | SHD | 1046.0 | 2470.0 | 4802.0 | 6626.0 | 9501.0 | 11931.0 | 14221.0 | 17349.0 | 20817.0 | 23673.0 | 45085.0 | 47184.0 |
|  | Runtime (s) | 6.1 | 9.5 | 14.9 | 24.6 | 49.9 | 74.8 | 140.2 | 219.8 | 371.6 | 684.6 | 7166.5 | 7229.8 |
| P/PC-Stable | F1 Adjacent | 75.1 | 73.2 | 68.1 | 68.1 | 64.9 | 64.8 | 62.9 | 61.3 | 59.3 | 58.6 | 50.6 | 32.5 |
|  | F1 Arrowhead | 46.6 | 45.9 | 39.1 | 38.0 | 35.9 | 35.2 | 34.2 | 32.6 | 31.7 | 30.7 | 26.1 | 16.8 |
|  | SHD | 899.0 | 1913.0 | 3512.0 | 4869.0 | 6670.0 | 8306.0 | 10163.0 | 12269.0 | 14830.0 | 16842.0 | 44491.0 | 54599.0 |
|  | Runtime (s) | 3.3 | 5.5 | 8.4 | 12.2 | 29.7 | 33.7 | 52.7 | 88.6 | 178.5 | 210.8 | 3649.3 | 6853.3 |
| P/fGES | F1 Adjacent | 69.5 | 59.5 | 51.9 | 50.4 | 46.2 | 43.6 | 42.1 | 39.5 | 35.9 | 34.1 | 5.9 | / |
|  | F1 Arrowhead | 62.3 | 49.2 | 42.1 | 41.3 | 38.2 | 35.5 | 35.0 | 32.9 | 29.0 | 28.3 | 4.8 | / |
|  | SHD | 1086.0 | 3361.0 | 6549.0 | 9357.0 | 13371.0 | 18043.0 | 21892.0 | 27495.0 | 36108.0 | 42805.0 | 30065.0 | 41037.0 |
|  | Runtime (s) | 4.8 | 20.4 | 60.3 | 131.7 | 406.7 | 604.0 | 801.4 | 1600.0 | 2585.5 | 3896.7 | 6549.9 | 6792.9 |
| PC-Stable | F1 Adjacent | 72.6 | 64.8 | 58.4 | 55.7 | / | / | / | / | / | / | / | / |
|  | F1 Arrowhead | 50.3 | 42.5 | 37.9 | 36.5 | / | / | / | / | / | / | / | / |
|  | SHD | 1261.0 | 3360.0 | 6192.0 | 9115.0 |  |  |  |  |  |  |  |  |
|  | Runtime (s) | 273.5 | 2997.3 | 19091.2 | 85197.4 | 86400.0 | 86400.0 | 86400.0 | 86400.0 | 86400.0 | 86400.0 | 86400.0 | 86400.0 |
| fGES | F1 Adjacent | 58.2 | 55.7 | 53.9 | 54.2 | 53.9 | 53.8 | 53.8 | 53.7 | 53.6 | / | / | / |
|  | F1 Arrowhead | 52.8 | 47.2 | 44.9 | 45.9 | 46.2 | 45.5 | 46.0 | 46.4 | 45.2 | / | / | / |
|  | SHD | 2107.0 | 4697.0 | 7364.0 | 9707.0 | 12123.0 | 14678.0 | 17045.0 | 19423.0 | 22128.0 |  |  |  |
|  | Runtime (s) | 600.9 | 2369.1 | 5453.8 | 10336.8 | 16623.9 | 26760.6 | 40465.7 | 60305.9 | 84889.9 | 86400.0 | 86400.0 | 86400.0 |

Table 9: Testing results on Asia problems with up to 30000 variables, in terms of F1 Adjacent, F1 Arrowhead, SHD, and runtime. On each network, the mean ± std performance obtained by each method on 10 problems is reported. The best performance in terms of accuracy metrics is marked with an underline, and the performance that is not significantly different from the best performance (according to a Wilcoxon signed-rank test with significance level $p = 0.05$) is indicated in **bold**.

| Problem ($|V|$, $|E|$) | | Asia-1000 (1000, 1100) | Asia-2000 (2000, 2200) | Asia-3000 (3000, 3300) | Asia-4000 (4000, 4400) | Asia-5000 (5000, 5500) | Asia-6000 (6000, 6600) | Asia-7000 (7000, 7700) | Asia-8000 (8000, 8800) | Asia-9000 (9000, 9900) | Asia-10000 (10000, 11000) | Asia-20000 (20000, 22000) | Asia-30000 (30000, 33000) |
|---|---|---|---|---|---|---|---|---|---|---|---|---|---|
| P/SLE | F1 Adjacent | **95.4** | **94.1** | **94.7** | **94.7** | **94.1** | **94.4** | **94.5** | **94.2** | **94.3** | **93.8** | **92.8** | **92.5** |
| | F1 Arrowhead | **73.7** | **72.6** | **73.0** | **73.4** | **72.4** | **72.8** | **72.9** | **72.6** | **72.8** | **71.8** | **71.1** | **70.9** |
| | SHD | **125.0** | **324.0** | **440.0** | **571.0** | **829.0** | **931.0** | **1061.0** | **1302.0** | **1449.0** | **1717.0** | **4103.0** | **6287.0** |
| | Runtime | 5.4 | 8.1 | 10.6 | 12.3 | 14.4 | 17.3 | 19.8 | 24.3 | 28.7 | 35.2 | 115.3 | 287.8 |
| P/SLE(D) | F1 Adjacent | 77.7 | 67.3 | 61 | 53.4 | 51.8 | 47 | 41.7 | 41 | 38.1 | 36.8 | 57.1 | 0.2 |
| | F1 Arrowhead | 62.9 | 55.2 | 50.8 | 43.8 | 42.4 | 38.4 | 34.3 | 33.4 | 31.2 | 29.9 | 24.3 | / |
| | SHD | 729 | 2307 | 4439 | 7890 | 10416 | 15021 | 21308 | 25178 | 31718 | 36831 | 30180 | 32963 |
| | Runtime | 12.4 | 24.8 | 52.8 | 141.6 | 250.9 | 524.2 | 984.1 | 1538.6 | 2210.7 | 3083 | 7230.6 | 8571.3 |
| P/SLE(R) | F1 Adjacent | 86.4 | 79.5 | 77.2 | 70.4 | 70.0 | 65.4 | 63.3 | 61.6 | 60.4 | 58.8 | 45.7 | 0.2 |
| | F1 Arrowhead | 48.1 | 40.4 | 37.5 | 30.9 | 31.5 | 29.1 | 27.5 | 26.2 | 26.2 | 25.1 | 19.4 | / |
| | SHD | 508.0 | 1577.0 | 2697.0 | 4832.0 | 6110.0 | 8688.0 | 11012.0 | 13366.0 | 15712.0 | 1844.0 | 45260.0 | 32962.0 |
| | Runtime | 5.0 | 7.1 | 10.2 | 16.9 | 24.2 | 51.4 | 91.5 | 141.2 | 214.5 | 374.8 | 7153.0 | 7220.1 |
| P/PC-Stable | F1 Adjacent | 90.5 | 86.1 | 85.2 | 81.5 | 80.0 | 77.5 | 75.0 | 73.9 | 72.9 | 71.4 | 61.0 | 35.1 |
| | F1 Arrowhead | 55.7 | 49.5 | 46.4 | 41.2 | 39.4 | 36.6 | 34.9 | 33.7 | 33.0 | 31.6 | 26.1 | 14.6 |
| | SHD | 352.0 | 1031.0 | 1700.0 | 2863.0 | 3859.0 | 5308.0 | 6889.0 | 8301.0 | 9745.0 | 1262.0 | 33245.0 | 41131.0 |
| | Runtime | 2.7 | 3.4 | 4.9 | 8.0 | 11.6 | 24.4 | 46.7 | 68.3 | 92.7 | 143.0 | 2521.7 | 7036.3 |
| P/fGES | F1 Adjacent | 77.7 | 67.3 | 61.0 | 53.4 | 51.8 | 47.0 | 41.7 | 41.0 | 38.1 | 36.8 | 8.2 | / |
| | F1 Arrowhead | 62.9 | 55.2 | 50.8 | 43.8 | 42.4 | 38.4 | 34.3 | 33.4 | 31.2 | 29.9 | 6.5 | / |
| | SHD | 729.0 | 2307.0 | 4439.0 | 7890.0 | 10416.0 | 15021.0 | 21308.0 | 25178.0 | 31718.0 | 36831.0 | 27119.0 | 33000.0 |
| | Runtime | 3.4 | 11.4 | 34.7 | 125.5 | 214.3 | 466.5 | 867.3 | 1370.8 | 1973.2 | 2838.6 | 6617.6 | 6629.0 |
| PC-Stable | F1 Adjacent | 73.1 | 62.5 | 56.8 | / | / | / | / | / | / | / | / | / |
| | F1 Arrowhead | 36.6 | 29.9 | 27.2 | / | / | / | / | / | / | / | / | / |
| | SHD | 1174.0 | 3457.0 | 6321.0 | | | | | | | | | |
| | Runtime | 992.5 | 9984.5 | 38956.1 | 86400.0 | 86400.0 | 86400.0 | 86400.0 | 86400.0 | 86400.0 | 86400.0 | 86400.0 | 86400.0 |
| fGES | F1 Adjacent | 50.5 | 48.5 | 48.0 | 47.9 | 47.6 | / | / | / | / | / | / | / |
| | F1 Arrowhead | 41.7 | 40.2 | 39.8 | 39.8 | 39.4 | / | / | / | / | / | / | / |
| | SHD | 2317.0 | 4982.0 | 7611.0 | 10208.0 | 12878.0 | | | | | | | |
| | Runtime | 2202.0 | 7762.0 | 19452.0 | 41567.9 | 68229.9 | 86400.0 | 86400.0 | 86400.0 | 86400.0 | 86400.0 | 86400.0 | 86400.0 |

