# OpenReview forum: "Auto-Ensemble Structure Learning of Large Gaussian Bayesian Networks"
_ICLR.cc/2025/Conference — ICLR 2025 Conference Withdrawn Submission_

### Official Review · Reviewer_q662 · 2024-10-25

**Soundness:** 3
**Presentation:** 2
**Contribution:** 2
**Rating:** 5
**Confidence:** 3

**Summary:**

This paper proposes Auto-Structure Learning Ensemble (Auto-SLE) for large-scale Gaussian Bayesian Network structure learning. The key innovation is automatically constructing an ensemble of structure learning algorithms to improve the stability and accuracy of learning across subproblems in large Bayesian networks. The proposed method integrates Auto SLE into the Partition Estimation Fusion (PEF) framework and achieves very good experimental results.

**Strengths:**

The paper proposes a novel and automated approach for ensemble-based structure learning in large Bayesian networks. Extensive experimentation shows significant improvements over state-of-the-art methods, with performance increases as high as 225% in terms of accuracy on large datasets.The proposed method generalizes well to different network sizes and characteristics, which increases its practical utility and theoretical guarantees for the performance of the Auto-SLE are provided, enhancing the paper's rigor.

**Weaknesses:**

The proposed Auto-SLE algorithm can essentially be seen as an improvement to the existing PEF algorithm, mainly addressing the instability in subproblem learning through an ensemble learning approach. As such, Auto-SLE is more of a stable version of PEF. However, the paper's description of the instability issue in PEF is relatively simple and lacks a deeper analysis of its root causes, which makes the innovation less prominent. Furthermore, the improvements in Auto-SLE are more of a refinement within the existing framework rather than offering a significant theoretical or methodological breakthrough, which somewhat diminishes the overall novelty of the work. In addition, a more detailed comparison with non-PEF-based methods in terms of their theoretical limitations and scalability would strengthen the paper's argument.

**Questions:**

1.The title mentions Gaussian Bayesian Networks (BN); however, there is insufficient description in the introduction regarding this aspect. Does the existing PEF method apply to general BNs, while the author's method is only suitable for Gaussian BNs? If the author's method applies to general BNs, why focus specifically on Gaussian BNs?
2.The title targets large-scale BN networks, but the introduction similarly lacks relevant description on this point.
3.Could the authors clarify whether Auto-SLE could be extended to non-Gaussian Bayesian networks or other types of graphical models?
4.What are the computational trade-offs involved when constructing the SLE automatically? Is there a limit to how large the configuration space can grow before performance deteriorates?
5.The proposed algorithm appears to be an improvement over the existing PEF algorithm, functioning as a more stable version of PEF. However, the shortcomings of the PEF method, specifically its instability, seem to be insufficiently described, which could suggest a lack of innovation in this work.

---

### Official Review · Reviewer_wLiq · 2024-10-27

**Soundness:** 3
**Presentation:** 3
**Contribution:** 2
**Rating:** 5
**Confidence:** 4

**Summary:**

This paper proposes an automatic method named Auto-SLE for selected near-optimal structure learning algorithms. Auto-SLE selects optimal structure learning algorithms by iteratively testing them with different hyperparameter sets. The authors theoretically prove the soundness of Auto-SLE and empirically evaluate its effectiveness on several public datasets. The results show that the algorithm selected by Auto-SLE, combined with PEF, outperforms other state-of-the-art structure learning algorithms in almost all scenarios.

**Strengths:**

1. The authors provide theoretical guarantees for their proposed Auto-SLE algorithm.
2. The experimental results seem to be a significant advance compared to other testing algorithms.

**Weaknesses:**

1. The theoretical guarantee for the test set needs to be clarified and should better be included in the appendix.
2. The experimental setting is not very fair for other structure learning algorithms since the Bayesian network for training Auto-SLE and testing are the same, which means Auto-SLE can use the ground truth to pick the optimal hyper-parameters for fGES and PC-stable, which will inevitably let PEF performs better than original fGES or PC-stable.
3. The evaluation of the performance of solely Auto-SLE is lacking.

**Questions:**

1. As mentioned in Section 3.4, “Importantly, when presented with a testing problem to solve, it cannot be assumed that the ground truth is known. Thus, quality measures that do not require ground truth, such as the BIC score adopted in our experiments, are used to select the best output from the outputs of member algorithms.” Could you please specify which experiment the BIC score was adopted as the quality measure?
2. Are the learned graphs converted to CPDAG before being assessed? Since linear Gaussian BNs with the same CPDAGs are statistically indistinguishable, comparing the learned CPDAG instead of the DAG would be more reasonable even when the algorithm returns a DAG.
3. How is the performance of P/SLE on learning the original BNs, i.e., Alarm, Asia, etc?

---

### Official Review · Reviewer_AVT9 · 2024-11-04

**Soundness:** 3
**Presentation:** 3
**Contribution:** 3
**Rating:** 6
**Confidence:** 4

**Summary:**

The paper proposed an approach of learning large-scale Bayesian network structures. The idea is based on divide-and-conquer. The authors proposed a pipeline to reach optimal combination of subproblems using greedy search approaches.

**Strengths:**

1. The idea is innovative on solving large-scale Bayesian network learning.
2. The experiment is extensive. Although some baseline algorithms are not recent, there are still representative.

**Weaknesses:**

1. The details of sub-problems solving and the relationship between the \theta configurations selected and the form of the final networks are not well described. It is hard for readers to figure out how the entire Bayesian network can be achieved using the proposed method.
2. Although there are not many BN learning methods proposed in recent years. The following recent methods using hybrid learning should be addressed and compared in some ways if possible:
(a) mFGS-BS, Chobtham et al. (2022),
(b) MAHC, Constantinou (2022)
(c) Kuipers J, Suter P, Moffa G (2022) J Comput Graph Stat 31:639–650

**Questions:**

How does the variable distributions affect your algorithm? Why you emphasis on Gaussian Bayesian networks? Can it applies to boolean/binomial variable settings?

---

### Official Review · Reviewer_R3aX · 2024-11-10

**Soundness:** 3
**Presentation:** 3
**Contribution:** 2
**Rating:** 3
**Confidence:** 3

**Summary:**

The paper focuses on problem of learning the structure of Bayesian networks (BNs) from data. It proposes a solution that enhances a recently proposed partition-estimation-fusion (PEF) algorithm for structure learning. In particular, the paper proposes to use structure learning ensemble (SLE) inside of the estimation step of PEF instead of doing it a single time. In order to find SLE, a greedy algorithm is proposed that adds algorithms to the ensemble one by one based on their performance of a set of training structure learning tasks. The proposed approach is evaluated on synthetic benchmark data.

**Strengths:**

+ The paper proposes an enhancement of a recently proposed algorithm
+ Within the experimental setting of this paper, the results show that the proposed algorithm provides a good tradeoff between runtime and quality of structure learning
+ The paper is easy to follow

**Weaknesses:**

- The idea to use structure learning ensemble within PEF algorithm is not particularly innovative. Thus, the merit of this work is not large.
- The paper enhances PFE, which is only one of the many structure learning algorithms. This, the impact of this work is not large
- Even though the paper provides experimental evaluation that is consistent with the related work, there is a fundamental issue with using Gaussian BN structure learning approaches on small-sample large-dimensional data. Evaluation in this paper goes into synthetic data with thousands or even tens of thousands of variables. The only justification to use BN structure learning is the hope to uncover causal relationships. When the number of variables is in hundreds or thousands and there is lack of prior knowledge related to causality in a particular domain, that hope is unrealistic. If the objective is predictive modeling or discovering covariance structure, there are more appropriate approaches than BN structure learning.

**Questions:**

The paper is quite clear. So, there are no questions.

---

### Note · Authors · 2024-11-19

I have read and agree with the venue's withdrawal policy on behalf of myself and my co-authors.